# Electrical Properties and Biological Synaptic Simulation of Ag/MXene/SiO$_2$/Pt RRAM Devices

**Xiaojuan Lian [1,2]**, **Xinyi Shen [1]**, **Jinke Fu [1]**, **Zhixuan Gao [1]**, **Xiang Wan [1]**, **Xiaoyan Liu [1]**, **Ertao Hu [1]**, **Jianguang Xu [3]** and **Yi Tong [1,*]**

[1] The Department of Microelectronics, Nanjing University of Posts and Telecommunications, Nanjing 210023, China; xjlian@njupt.edu.cn (X.L.); 1218023032@njupt.edu.cn (X.S.); 1220024213@njupt.edu.cn (J.F.); 1220024217@njupt.edu.cn (Z.G.); wanxiang@njupt.edu.cn (X.W.); xiaoyanliu@njupt.edu.cn (X.L.); iamethu@njupt.edu.cn (E.H.)

[2] National and Local Joint Engineering Laboratory of RF Integration and Micro-Assembly Technology, Nanjing 210023, China

[3] The School of Materials Science and Engineering, Yancheng Institute of Technology, 211 East Jianjun Road, Yancheng 224051, China; jgxu@ycit.cn

\* Correspondence: tongyi@njupt.edu.cn

**Abstract:** Utilizing electronic devices to emulate biological synapses for the construction of artificial neural networks has provided a feasible research approach for the future development of artificial intelligence systems. Until now, different kinds of electronic devices have been proposed in the realization of biological synapse functions. However, the device stability and the power consumption are major challenges for future industrialization applications. Herein, an electronic synapse of MXene/SiO$_2$ structure-based resistive random-access memory (RRAM) devices has been designed and fabricated by taking advantage of the desirable properties of SiO$_2$ and 2D MXene material. The proposed RRAM devices, Ag/MXene/SiO$_2$/Pt, exhibit the resistance switching characteristics where both the volatile and nonvolatile behaviors coexist in a single device. These intriguing features of the Ag/MXene/SiO$_2$/Pt devices make them more applicable for emulating biological synaptic plasticity. Additionally, the conductive mechanisms of the Ag/MXene/SiO$_2$/Pt RRAM devices have been discussed on the basis of our experimental results.

**Keywords:** RRAM devices; 2D MXene; resistance switching; volatile; nonvolatile; synaptic plasticity

---

## 1. Introduction

Brain-inspired computing systems have been extensively investigated for their capability to break through the bottlenecks of dominant von Neumann computer architecture [1–8]. Different kinds of electronic synapses, the key components of brain-inspired systems, have been proposed to imitate biological synaptic functions, including transistors [9,10], phase change memory [11,12], ferroelectric devices [13,14], and resistive random-access memory (RRAM) devices [6–8], among others. Specifically, RRAM devices are considered one of the most competitive candidates as electronic synapses due to their low power consumption, high speed switching, multiple resistance states and so on [15–25]. However, the device reliability of the RRAM devices hinders the commercialization in future machine learning and neuromorphic computing and the power consumption requires further reduction. To break through these challenges, new materials and innovative structure is demanded.

Two-dimensional (2D) material MXene, a family of transition metal carbides or nitrides, has attracted considerable attention in different research fields for their layered structure, high stacking density, ultra-high conductivity (~6000–8000 S/cm), fast charge response and other

outstanding performances [26–35]. Generally, MXene is in the form of $M_{n+1}X_n$ where M is early translation metal and X is carbon or nitrogen. MXene can be synthesized by wet etching A from MAX compounds (A is Al or Si) due to the stronger connection of M-X bond than M-A bond [29,30]. Moreover, MXene can become narrow-band gap semiconductors or metal due to surface functionalities such as -O, -F, or -OH [31]. It has been reported that some 2D materials such as Graphene, $MoS_2$, BN, and $WS_2$ [17,36–39] are able to ameliorate the performances of RRAM devices. Therefore, by combining the compatibility of $SiO_2$-based RRAM devices with the dominant silicon CMOS fabrication technology [40–42] and those of a 2D MXene material, we designed and fabricated Ag/MXene/$SiO_2$/Pt structure-based RRAM devices. The introduction of 2D material MXene greatly lowers the operation voltage of silicon dioxide RRAM devices by controlling the growth of conductive filaments (CFs). In addition, the studied devices exhibit the resistance switching (RS) characteristics where both the volatile and nonvolatile behaviors coexist in a single device. The coexistence of the volatile and nonvolatile switching, capable of implementing short-term plasticity (STP) and long-term plasticity (LTP) rules, respectively, can effectively reduce the complexity of brain-inspired systems [43]. Finally, the working mechanisms of the Ag/MXene/$SiO_2$/Pt devices have been discussed on the basis of our experimental results. This work may provide a forward-looking solution for reducing the power consumption of traditional transition metal oxide-based RRAM devices as well as the development of artificial intelligence systems on the hardware level.

## 2. Materials and Methods

The studied Ag/MXene/$SiO_2$/Pt RRAM devices were successfully fabricated on Si wafer. Figure 1a shows the scanning electron microscope (SEM) image of the crossbar structure of devices that are more likely to be integrated and extended with the conventional silicon CMOS fabrication technology. The vertical lines of the array are bottom electrodes of 90-nm-thick Pt and the parallel lines are top electrodes of TiN/Ag (80 nm/100 nm). Pads (300 μm × 300 μm) are illustrated by white dashed lines. The red dashed line frames the core intersection region (100 μm × 100 μm) of top and bottom electrodes, where the RS occurs. The 80 nm RS layer of $SiO_2$ film was sputtered using the physical vapor deposition (PVD) method. The MXene ($Ti_3C_2$) layer on $SiO_2$, prepared by etching $Ti_3AlC_2$ with hydrogen fluoride (HF), was deposited by spin-coating at 500 rpm for 60 s. The cross-sectional SEM image indicates that the thickness of the MXene film is about 50 nm, as shown in Figure S1. The surface morphology of $Ti_3C_2$ powder was characterized by SEM (Figure 1b) showing the chip-like multi-layered nanostructure. The X-ray diffraction (XRD) figure indicates that the main component of the MXene used in our experiment is $Ti_3C_2$ [44–46], as shown in Figure 1c. All the electrical characteristic measurements were performed by Keithley 4200A SCS semiconductor parameter analyzer.

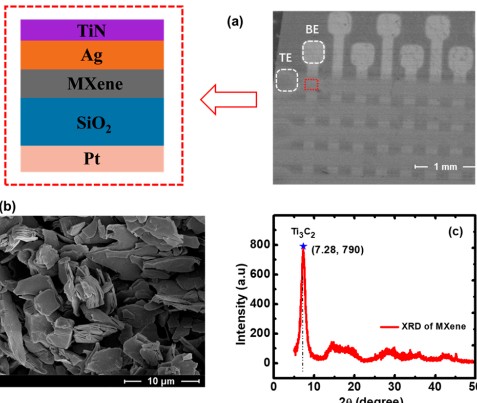

**Figure 1.** (**a**) Scanning electron microscope (SEM) image of the TiN/Ag/MXene/$SiO_2$/Pt crossbar structure. (**b**) The surface morphology of $Ti_3C_2$ powder was characterized by SEM, showing the classical multi-layered nanostructure. (**c**) The X-ray diffraction (XRD) shows that the MXene used in this experiment is mainly composed by $Ti_3C_2$.

## 3. Results

### 3.1. Electrical Characteristics

By controlling a compliance current limit, we can obtain the electrical characteristics in Ag/MXene/SiO$_2$/Pt RRAM devices where both volatile and nonvolatile behaviors coexist in a single device. The volatile switching behavior would appear when the compliance current limit is relatively low (i.e., approximately from 1 to 500 nA), whereas the nonvolatile switching behavior occurs with a relatively high compliance current limit (i.e., approximately from 10 μA to 5 mA). Cycling experiments were performed on the Ag/MXene/SiO$_2$/Pt RRAM devices under different compliance current limits, each of which consists of 100 cycles. Figure 2a,b shows the typical volatile *I-V* characteristic curves under the compliance current limits of 1 and 100 nA, respectively. They have similar threshold switching (TS) behaviors [47], wherein the forward scanning (from 0 to 0.2 V) excites them into the low resistance state (LRS) and the reverse scanning (from 0.2 to 0 V) makes them completely return to the high resistance state (HRS). Without the Reset operation, the device would automatically return to the HRS. Furthermore, we statistically analyzed the Set voltages of 100 cycles under two different compliance current limits, as shown in Figure 2c,d. Most values of Set voltages are around 0.06 V for a compliance current limit of 1 nA, while the values of Set voltages are observed around 0.18 V for a higher compliance current limit of 100 nA. Irrespective of this, they are both ultra-low Set voltages applicable to low-power artificial neural microcircuits. It should be noted that Set voltages decrease with cycles increasing under a relatively low compliance current limit (approximately from 1 nA to 500 nA), which may be attributed to few residual metallic Ag nanoparticles during the former phase [48] and make the device easier to turn on at lower operation voltages. More supporting materials for TS behaviors can be found in supplementary Figures S2–S4.

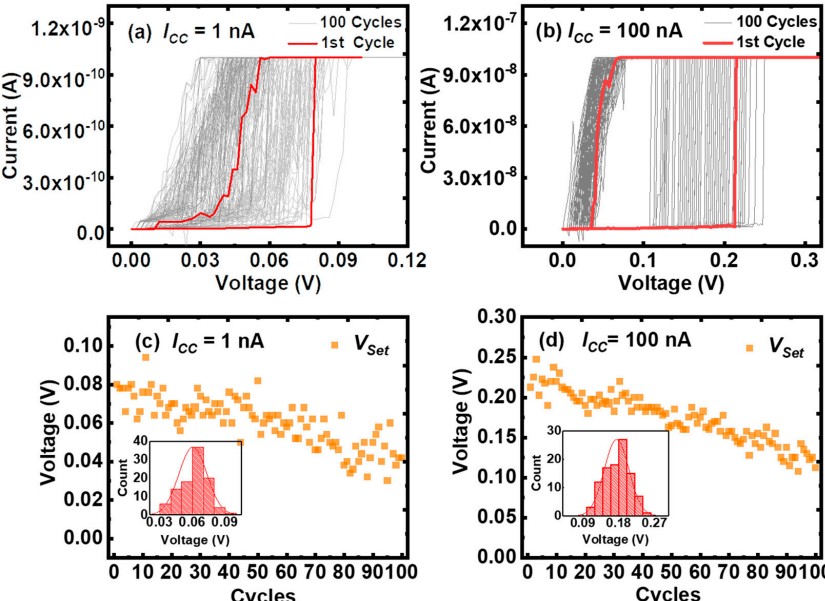

**Figure 2.** Typical threshold switching (TS) *I-V* curves under the compliance current limits of (**a**) 1 and (**b**) 100 nA. The statistics and distribution of Set voltages under the compliance current limits of (**c**) 1 nA, most of values are around 0.06 V, and (**d**) 100 nA, the values of Set voltages are observed around 0.18 V.

When we continuously increased the value of the compliance current limit, non-volatile behavior was achieved in the same Ag/MXene/SiO$_2$/Pt device. Figure 3a,b shows the typical RS *I-V* characteristics curves of 100 cycles under the compliance current limits of 500 μA and 1 mA, respectively. They have similar RS behaviors, wherein the Set process puts them into the LRS under positive voltage sweep and the Reset process restores them back to the HRS under the negative voltage sweep. This process

is completely different from volatile behavior that can automatically return to the HRS without the Reset operation. In addition, the resistances of ON and OFF states at 0.01 V were extracted for both compliance current limits of 500 μA and 1 mA, as shown in Figure 3c, d. It can be observed that the resistance ratio of the OFF-State and ON-State is almost $10^3$. Besides, the data retention in both ON and OFF states was measured under the compliance current limits of 100 μA (shown in Figure S5), 500 μA and 1 mA, respectively. From Figure 3e,f, we can see that a relatively high and stable retention (more than $1 \times 10^4$ s) can be obtained under a higher compliance current limit of 1 mA, which means that the stronger the electrical stimulation, the longer the data retention. This phenomenon is consistent with the LTP characteristic of biological synapses, which is regarded as the basis of learning and memory [20]. However, the Set and Reset voltages in these two RS processes are still very low, around 0.2 V and −0.2 V, respectively, benefiting the development of low-power brain-inspired computing systems. The low operating voltages in both TS and RS situations are more likely to be associated with the introduction of 2D materials MXene, which makes the CFs tend to grow along the location of MXene nanostructures. Our recent research work has demonstrated that the $Ti_3C_2$ appears to be bonded to the amorphous $SiO_2$ to form an MXene/$SiO_2$ compound due to the surface activity of the dangling bonds in the cleaved amorphous $SiO_2$ substrate [33].

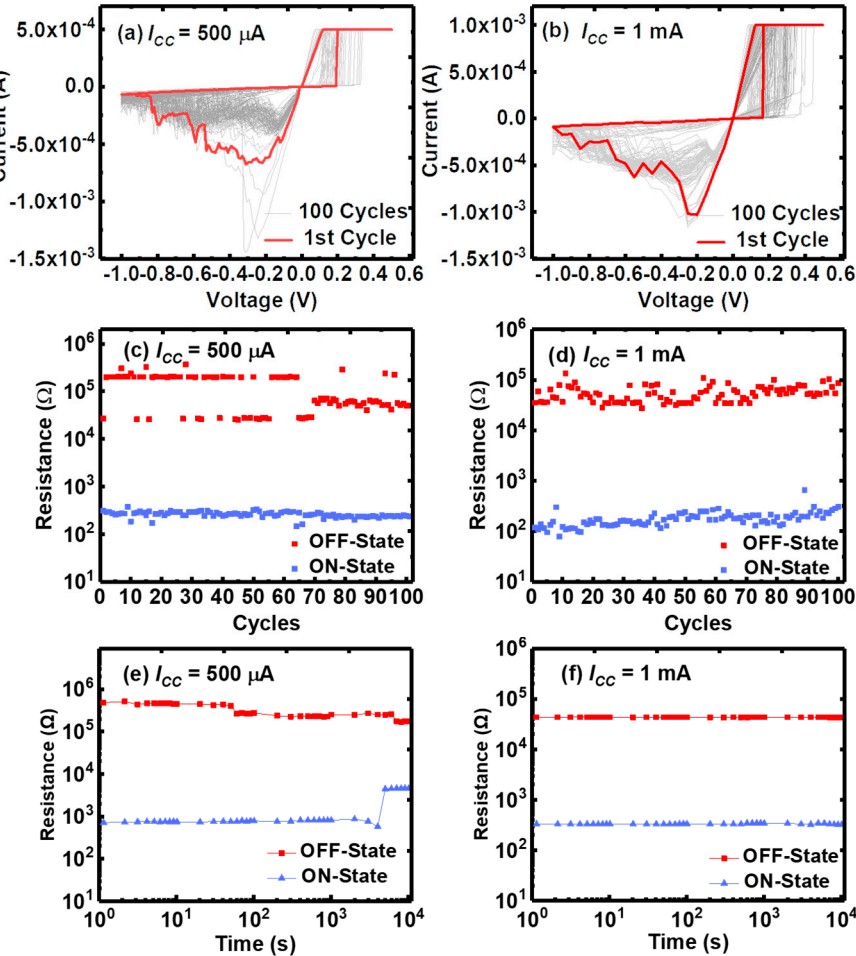

**Figure 3.** Typical resistance switching (RS) *I-V* curves of 100 consecutive cycles (gray curves) under the compliance current limits of (**a**) 500 μA and (**b**) 1 mA, respectively. ON and OFF resistance states during 100 cycles under the compliance current limit of (**c**) 500 μA and (**d**) 1 mA, respectively, extracted at 0.01 V. The data retention under the compliance current limits of (**e**) 500 μA and (**f**) 1 mA for the same device.

### 3.2. Biological Synaptic Simulation

In biological synapses, once the action potential (AP) arrives at the presynaptic nerve terminal, excitatory postsynaptic current (EPSC) would generate with the open of ion channels of $Na^+$ in postsynaptic membranes, which causes the change of synaptic weight between two neurons, as shown in Figure 4a [4,5]. As a pair of identical pulses is given to the presynaptic nerve terminal, EPSC-2 caused by the second pulse is significantly enhanced. This phenomenon is called paired-pulse facilitation (PPF) of synapses, an important characteristic of STP [49,50]. Hereby, we simulated the PPF characteristic in Ag/MXene/SiO$_2$/Pt RRAM devices. A pair of spikes with the pulse amplitude of 0.2 V and pulse width of 90 ms was applied to the Ag/MXene/SiO$_2$/Pt device, and the corresponding responses of the PPF characteristic are obtained, as shown in Figure 4b. It can be observed that EPSC-2 is much larger than EPSC-1 caused by the first pulse, which is consistent with the phenomenon of biological synaptic plasticity [51,52]. The PPF characteristic can be better elucidated by considering the change between two peak currents of EPSC-2 and EPSC-1 induced by a pair of identical pulses versus the interval time ($\Delta t$). As shown in Figure 4c, the index of $PPF = (I_2 - I_1)/I_1 \times 100\%$ exponentially decreases with $\Delta t$ increasing, $I_1$ and $I_2$ are the values of the peak currents after the first and second pulses, respectively. In this experiment, the index of PPF can be fitted with the following equation:

$$PPF = C_1 \exp(-\Delta t/\tau_1) + C_2 \exp(-\Delta t/\tau_2)$$

where two characteristic relaxation times $\tau_1 = 5.83$ ms and $\tau_2 = 73.45$ ms were obtained, corresponding to fast and slow decay terms, respectively [20,53]. The constants of $C_1$ and $C_2$ are equal to 1.21 and 694.98, respectively. The PPF characteristic in the Reset process was also obtained and the index of PPF exponentially decreases with $\Delta t$ increasing as well (Figure S6). Furthermore, the LTP characteristic has been mimicked in our Ag/MXene/SiO$_2$/Pt RRAM devices, as shown in Figure 4d. After applying 100 consecutive positive spikes (pulse amplitude is 0.2 V, pulse width is 25 ms), the conductance of devices begins to gradually increase from the initial state to a saturation resistance state. This process lasts for a long period of time and is termed the long-term potentiation (red scatters). Subsequently, 100 consecutive negative spikes (pulse amplitude is 0.2 V, pulse width is 25 ms) were applied to the same device. The conductance gradually decreases to a certain resistance state, being known as the long-term depression (blue scatters). Commonly, the LTP is regarded as the basis of learning and memory [20,54–56] that can be applied into the synaptic weights training in artificial neural networks [4].

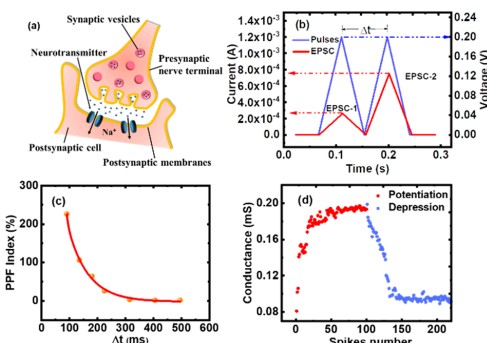

**Figure 4.** (**a**) The structural schematic of the signal transmission between synapses. (**b**) The blue curve is a pair of identical spikes applied to the top electrode with the pulse amplitude of 0.2 V and pulse width of 90 ms. The red curve is the corresponding responses of the paired-pulse facilitation (PPF) characteristic, and the excitatory postsynaptic current (EPSC)-2 is significantly enhanced. (**c**) The relationship between the index of PPF and pulse intervals $\Delta t$, displaying an exponential decrease. (**d**) The long-term potentiation (red scatters) and long-term depression (blue scatters) characteristics of the Ag/MXene/SiO$_2$/Pt RRAM devices.

## 4. Discussion

The Ag/MXene/SiO$_2$/Pt RRAM devices have been designed and fabricated for the application of electronic synapses. By controlling a compliance current limit, the RS characteristics can be obtained in a single Ag/MXene/SiO$_2$/Pt RRAM device where the volatile and nonvolatile switching behaviors coexist. The volatile switching behavior appears when the compliance current limit is relatively low, whereas the nonvolatile switching behavior occurs with a relatively high compliance current limit. Furthermore, the synaptic plasticity functions have been mimicked in the same Ag/MXene/SiO$_2$/Pt RRAM device, such as the PPF and LTP characteristic.

After combining the electrical characteristics with the biological synaptic simulation, we speculated the RS mechanism. When the positive voltage is applied to the Ag/MXene/SiO$_2$/Pt RRAM device, the Ag ions generate by oxidation reaction at the top electrode and migrate to the bottom electrode. The metallic CFs of Ag would form with the accumulated Ag atoms, and this is the Set process. When a relatively low compliance current limit was applied to the Ag/MXene/SiO$_2$/Pt device, a narrow metallic CF may be composed of discrete Ag nanoparticles [48,57,58] and forms in the RS layer. The CFs in this situation are unstable and tend to rupture spontaneously. The above process is the TS: the device would automatically return to the HRS without the Reset operation, similar to the STP characteristic of biological synapses. When we apply a relatively high compliance current limit, a thicker CF would be formed in the Ag/MXene/SiO$_2$/Pt RRAM device. In this situation, the ON-State can be maintained after the Set operation and the Reset process is needed to restore them back to the OFF-State again. This is the typical nonvolatile RS process that is consistent with the LTP characteristic of biological synapses. Moreover, it is noted that the Ti$_3$C$_2$ appears to be bonded to the amorphous SiO$_2$ to form an MXene/SiO$_2$ compound due to the surface activity of the dangling bonds in the cleaved amorphous SiO$_2$ substrate [33], which makes the CFs more likely to grow along the location of MXene nanostructures, accordingly reducing the power consumption of Ag/MXene/SiO$_2$/Pt RRAM devices. Finally, the electrical characteristics of the Cu/MXene/SiO$_2$/W RRAM device under the compliance current limits of 500 μA and 1 mA (shown in Figure S7) further demonstrate our proposed RS mechanism where the RS processes are determined by oxidation-reduction reactions of Cu or Ag ions in the SiO$_2$ layer, which rule out the phase transition characteristic of the MXene material.

**Supplementary Materials:** The following are available online at http://www.mdpi.com/2079-9292/9/12/2098/s1, Figure S1: a single layer or few-layered MXene on glass slide and the cross-sectional SEM image of the MXene film, Figure S2: the forming processes under different compliance current limits, Figure S3: the TS behaviors in both positive and negative bias directions under the compliance current limit of 100 nA, Figure S4: the statistics of TS voltages versus the initial HRS resistances under the compliance current limits of 1 nA and 100 nA, respectively, Figure S5: the retention measurement under the compliance current limit of 100 μA, Figure S6: the PPF characteristic in the Reset process, Figure S7: The typical *I-V* curves of Cu/MXene/SiO$_2$/W RRAM device under the compliance current limits of 500 uA and 1 mA, respectively.

**Author Contributions:** Conceptualization, X.L. (Xiaojuan Lian) and Y.T.; methodology, X.L. (Xiaojuan Lian), X.S., E.H.; investigation, X.L. (Xiaojuan Lian), X.S. and X.W.; data curation, X.S., J.F. and Z.G.; writing—original draft preparation, X.L. (Xiaojuan Lian) and X.S.; writing—review and editing, X.L. (Xiaojuan Lian), X.S., J.F., Z.G., X.W., X.L.(Xiaoyan Liu), E.H., J.X. and Y.T.; supervision, X.L. (Xiaojuan Lian); project administration, X.L. (Xiaojuan Lian) and Y.T.; funding acquisition, X.L. (Xiaojuan Lian), J.X. and Y.T. All authors have read and agreed to the published version of the manuscript.

**Funding:** This research was funded in part by the National Natural Science Foundation of China (61804079, 21671167), the Science Foundation of Jiangsu Province (CZ1060619001, SZDG2018007), the Science Foundation of National and Local Joint Engineering Laboratory of RF Integration and Micro-Assembly Technology (KFJJ20200102), the Science Research Funds for Nanjing University of Posts and Telecommunications (NY218110, BK20191202).

**Conflicts of Interest:** The authors declare no conflict of interest.

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
