# Peer review of "Electrical Properties and Biological Synaptic Simulation of Ag/MXene/SiO2/Pt RRAM Devices"

_electronics, doi:10.3390/electronics9122098_

Round 1

Reviewer 1 Report

The manuscript (electronics-1013392), Electrical Properties and Biological Synaptic Simulation of Ag/MXene/SiO2/Pt RRAM devices, shows the results of MXene-based RRAM device and potential synaptic device applications. Authors present quite comprehensive analysis electrically and physically. However, a lot of technical work still need to be address before further confirm the outcomes for this referee, comments shown as following - 

  1. In Fig. 1 (b), the SEM images don't present the 2D structure in MXene, and also, what is the thickness layer information for this MXene? It should be related to the "Bulk" device region rather than as claimed 2D material. Please provide the TEM images to confirm the layer thickness information in device structure. 
  2. A lot of supporting data needs to be provided here to clarify the MXene layer properties. Are they semiconductor-type or metal-type properties? What is the conductivity in this layer? How to rule out the potential phase change characterization in MXene since the XRD results showed that MXene is crystalline. In Page 2, authors claim that the MXene is Ti3AlC2 but later on become the Ti3C2, where the Al composition?
  3. In Fig. 2, can authors provide the forming process and statistical data in manuscript? For the volatile threshold switching behaviors, does the forming process also need to set up a lower compliance current to activate the volatile switching behaviors? Also, for the volatile switching, is that unipolar or nonpolar? Does the negative bias direction also have threshold switching behaviors? Why the compliance current limitation setting would impact the threshold switching voltage? Please provide the cycling-to-cycling and device-to-device HRS resistance data to support here. And do any correlation authors can find between the HRS resistance values and threshold switching? Or no correlation? Why?
  4. In Fig. 3, can authors shown the one more compliance current (with lower compliance current) data to support the retention data trend? Did we still find out both HRS and LRS degradation or simply LRS compliance degradation? And why the 500uA HRS retention degradation more severely as compared to 1mA? The compliance current limitation should not impact the HRS retention or it characteristics? More explanation needed at this point.
  5. In Fig. 4 (d), it should include both LTP and LTD behaviors. Also, in Fig. 4 (b) and Fig. 4 (c), it looks like authors perform the PPF on the positive SET region, can authors perform the same experimental setting in the RESET process? Did the authors find out the difference between the SET-PPF vs. RESET-PPF? Why?
  6. For the RS mechanism discussion, please provide the carrier transport behaviors study in detail for both HRS and LRS. Also, since a lot of research work found out that purely SiOx layer can also have the resistive switching characteristics (Journal of Applied Physics 116 (4), 043709 (2014), Journal of Applied Physics 116 (4), 043708 (2014), Journal of Applied Physics 112 (12), 123702 (2012), Applied Physics Letters 103 (19), 193508 (2013)), how to rule out the resistive switching not driven by the SiOx here?

Due to the above comments, this referee would like to put the manuscript status as "Major Revision" in the current phase.

Author Response

Please see the attachment, thank you!

Reviewer 2 Report

The authors presents electrical properties and biological synaptic simulation of Ag/MXene/SiO2/Pt RRAM devices. The idea is well conceived and the research is appropriately designed. Also, the paper is well written. However, there are some minor concerns that need to be addressed before publication.

1- The introduction is incomplete. follow the MDPI instructions to write the introduction section. explain the novelty and the importance of this study and the most important results in the last paragraph of introduction section.

2- Provide the jcpds card number with witch the xrd peaks are confirmed? 

3- In the end, provide the limitations to this study and the future directions?

Author Response

Please see the attachment, thank you!

Round 2

Reviewer 1 Report

This is the 2nd review process. Although authors replied to this referee in detail, however, some points still need to be further addressed below - 

  1. If authors cannot provide the TEM images, please provide the AFM to confirm the thickness information for both SiO2 and MXene. Also, what is the composition ratio in SiOx layer? That would also quite impact on the CBRAM or RRAM characterization by much. Please characterize here by XPS or EDX. 
  2. Authors mentioned that "in our work, MXene is considered to be semiconductor-type", did we know what type of semiconductor authors mentioned here? N-type or P-type, and what is the energy bank gap for the layer stacked MXene here? Does any semiconductor properties difference between the single layer MXene and stacked layer MXene? Also, authors did not answer this question - How to rule out the potential phase change characterization in MXene since the XRD results showed that MXene is crystalline.
  3. Please revise the Fig. 4 (d) Figure caption. Also, please add the RESET-PPF (overlap with Fig. 4 (c)) and discuss what is the difference between the SET-PPF and RESET-PPF. 
  4. Along with Q6, the OPC fitting cannot fully represent the CBRAM characterization, which is the potential mechanism proposal in this work; in-situ study also did not provide in this work to further confirm the Ag filament here. Authors should provide more detailed discussion (e.g. change the Ag electrode to other electrode materials, did we still see the resistive switching behaviors) to support the resistive switching mechanism discussion in the manuscript (this should also help to answer the Q2 above for the phase change possibility rule out). 
  5. Please add the materials replied in the response letter (rev 1 and rev 2) into supporting information attached with the original manuscript for this referee and potential readers reference.  

Due to the above comments, this referee still put the status as "Major Revision" in the current phase. 

Author Response

Please see the attachment,thank you.

Round 3

Reviewer 1 Report

Authors have replied to this referee in detail. No additional comments from this referee.